# The Use of Optical Coherence Tomography and Electrophysiological Tests in the Early Diagnosis of Inflammatory Changes in the CNS in children with ASD—A Review of Contemporary Literature

**DOI:** 10.3390/ijerph20043591

**Published:** 2023-02-17

**Authors:** Monika Modrzejewska, Wiktoria Bosy-Gąsior

**Affiliations:** 12nd Department of Ophthalmology, Pomeranian Medical University in Szczecin, Powstańców Wielkopolskich 72, 70-111 Szczecin, Poland; 2Scientific Association of Students 2nd Department of Ophthalmology, Pomeranian Medical University in Szczecin, Powstańców Wielkopolskich 72, 70-111 Szczecin, Poland

**Keywords:** autism spectrum disorder (ASD), ophthalmic manifestations of visual organ, risk factors, OCT, ERG

## Abstract

This article is a review of the contemporary literature on the possibility of using modern ophthalmological diagnostics, such as optical coherence tomography and electrophysiological tests, in the assessment of changes in eyesight correlating with inflammatory changes in the central nervous system (CNS) as one of the risk factors for neurodevelopmental disorders in children with ASD. A significant role is attributed to the activation of nerve and glial cells, as well as inflammatory changes in the brain, both of which can be of great importance in regard to an autism development predisposition. This fact indicates the possibility of using certain ophthalmic markers to depict an early correlation between the CNS and its outermost layer, i.e., the retina. A comprehensive ophthalmological assessment, and above all, characteristic changes in the functional function of photoreceptors and disorders of the structures of the retina or optic nerve fibers found in the latest OCT or ERG tests may in the future become diagnostic tools, further confirming the early characteristics of autism in children and adolescents. The above information, therefore, emphasizes the importance of cooperation between specialists in improving the diagnosis and treatment of children with autism.

## 1. Introduction

ASD illnesses are classified as neurodevelopment diseases and are characterized by persistent difficulties in social communication and interaction, as well as limited and repetitive behavioral patterns, interests, and activity [1]. Depending on the region, the prevalence of autism ranges from 0.2% in Europe [2,3] and Asia [4,5], up to 1.85–2.5% in the USA [6,7], 2.5% in Australia [8], and 3.13% in Iceland [9]. The difference between these regions is explained by varying amounts of published epidemiological studies concerning the problem. Within the last decade, the incidence of ASD diagnoses has increased by 0.7% (before: 0.83%, currently: 1.53%) [10]. The disorder is more frequently found among boys than girls (4.4:1 ratio) [11].

A review of the literature from 2017 to 2023 found in the PubMed and Google Scholar search engines that many papers note the correlation between autistic disorders and pre- and perinatal factors. From a pathophysiological standpoint, the influence of cytokines, IL-17A and IL-2, connected to encephalitis, is described [12,13]. In addition, IL-6 is named as a significant marker in the development of autism [14,15]. Some theories point to viral intrauterine (such as rubella, measles, mumps, cytomegalovirus (CMV), polyomavirus, and influenza), bacterial (*Mycoplasma ssp.*, *Chlamydia pneumoniae*, and *Borrelia burgdorferi*), and parasitic infections, especially those of the TORCH group, as the ones playing a notable part in the development of ASD [14,15,16]. Other factors of the perinatal period begging concern are birth complications, prolonged labor, the oxidative stress in the child’s body that accompanies it, birth asphyxia, and prematurity [17]. Among children with ASD, ophthalmologic symptoms occur with an incidence of 26.9–40% [18,19]. It was found that the symptomology in this group of patients is much more severe and complex than among children developing properly [20], which is mainly due to difficulties in cooperation and communication between the autistic patient and the ophthalmologist. Thus, changes in the organ of vision can be overlooked, which can cause improper motor development and a lack of improvement in the functioning of a child with ASD [18].

Due to the limited amount of articles on visual impairment among children with autism (PubMed search engine: 74 articles; key words: visual impairment, autism, and ophthalmology), the authors shall present the association between the state of the organ of vision and ASD, extending the current published pool of knowledge by the neurological risk factors influencing the development of autism (PubMed search engine: 205 articles). Moreover, in the available world literature, so far there are no reports describing the usefulness of ophthalmological tests such as OCT and ERG as tools for early diagnosis of visual disorders in children with ASD—hence the interest in this topic by the authors of this article.

## 2. ASD Biological Substrate

From a biological perspective, the development of ASD is found to be correlated with, among others, an increased rate of growth of the amygdala between the 6th and 12th month of life, which greatly precedes the appearance of social deficiencies and a final diagnosis of autism itself. Early improprieties in the growth of the amygdala occur at the same time as the development of the surface of the visual cortex and, simultaneously, as the observance of sensorimotor as well as attention deficit disorders among infants, in which ASD will have developed in the future [21]. 

Likewise, one observes dysfunctions related to the processing and integration of sensory information in the cerebellum, cerebral cortex, other parts of the limbic system (especially the hippocampus), corpus callosum, basal ganglia, and brain stem as well as changes in the levels of secretion of neurotransmitters found in the structures listed above, which indirectly influence the emergence of visual impairments [20]. The latest reports show thorough proof of glutaminergic dysfunction—changed expression, transport, and function in both metabotropic (i.e., glutaminergic) and ionotropic (i.e., NMDA) receptors, which results in a modified plasticity and synaptic development in children with ASD. Knowledge in this domain could prove useful in the application of pharmacotherapy for this group of patients [22]. A dysregulation at the transmitter level, i.e., acetylcholine, dopamine, and melatonin, is also mentioned. It is known that acetylcholine plays an important role in the pre- and postnatal neurodevelopmental processes (also in cell survival), whereas the early activity of dopamine and melatonin remains unknown. In further stages, the level of these substances correlates with circadian rhythm disorders [23].

A significant role is also attributed to the activation of nerve and glial cells and inflammatory changes in the brain, which can be of great importance in regard to predispositions to the development of autism (in the presence of other external factors). The aforementioned factors have both polygenic and environmental backgrounds and, most essentially, can be used in both experimental and clinical studies on treating autism [24].

## 3. Congenital and Perinatal Factors

### Perinatal Disorders and CNS Damage

Most perinatal disorders correlated with the development of ASD mainly involve pathologies that are developed within the central nervous system. Studies show that about 5–46% of children diagnosed with ASD struggle with the aforementioned epilepsy, and 30% of children with epilepsy meet the criteria for ASD [25]. There are theories that the predisposition to the co-occurrence of epilepsy and ASD shows common biological pathways ranging from gene transcription to altered synaptic function. SCN1A (14.9%) and MECP2 (10.6%) were identified as the most common genetic mutations [26]. The occurrence of epilepsy closely correlates with perinatal life complications in newborns (observed in premature infants), in particular with brain damage, such as diagnosed intraventricular hemorrhage (IVH) [27]. According to a study by Movsas et al., IVH is also a high-risk factor for the development of autism in a child [28]. The occurrence of perinatal hypoxia is mentioned as one of the factors in the development of ASD. A study by Maramara et al. found a significant correlation (*p* < 0.001) [29], while a study by van Tilborg et al. (performed on mice) observed an association between autism-like behavioral disorders and postnatal hypoxia—present in animals from pregnancies complicated by intrauterine infection [30]. 

Infection as a factor in the development of various diseases of prematurity, such as retinopathy of prematurity (ROP) [31], is also a cited factor that increases the risk of developing autism—as evidenced by a meta-analysis by Tioleco et al. A clear decrease in the incidence of ASD is observed when infections are accurately diagnosed and promptly treated during pregnancy [32]. The aforementioned researchers show a lack of correlation between specific pathogens and the trimester during which the onset of infection takes place [32], which the study by Ornoy et al. disagrees with, clearly separating the concept of prenatal infections and indicating some correlation between the rubella virus and CMV (i.e., TORCH pathogens) and a possible correlation with the influenza virus and toxoplasmosis. However, it excludes other maternal infections due to viruses, i.e., HPV, EBV, or Parvovirus [33]. 

A confirmation of the high involvement of immune factors is provided by studies of immune abnormalities against brain tissue antigens in autism. Mention is made, among others, of dysfunctions within the cell lines of T lymphocytes, B lymphocytes (including production of immunoglobulins), NK cells, and in the increased production of pro-inflammatory cytokines. It has been suggested that the occurrence of prenatal viral infections can lead to damage to the immature immune system and can induce viral tolerance, as well as T-cell activation with the subsequent development of an autoimmune response [24].

## 4. Ophthalmic Manifestations

### 4.1. Refractive Errors, Strabismus, and Astigmatism 

The incidences of changes in the organ of vision among patients with ASD have been found to be 26.9% in the study by Kabatas et al. and Black et al. [18,34], 33% in the study by Ezegwui et al. [35], and 40% in the study by Ikea et al., where 29% thereof had a significant refractive error, most of which was hyperopia (16.9%), then high myopia (5.8%), high astigmatism (3.9%), and anisometropia (1.9%) [19]. The study by Kabatas et al. showed significant refractive errors among 22% of patients, where hyperopia constituted the largest share (8.3%), then simple astigmatism: hyperopic (4%), myopic (3.4%), and compound astigmatism: hyperopic (2.5%), myopic (1.5%), and myopia (1.9%) [18]. Similarly to the authors above, Denis et al. also found hyperopia to be the most commonly occurring error among children with ASD with 70% incidence, astigmatism > 1 diopter among 60%, astigmatism (mainly in the oblique axis)—bilateral in 40% and unilateral in 20% [36]. 

Kabatas et al. diagnosed strabismus in 8.6% of children [8]; Ikeda et al. in 41% [9]; and Denis et al. in 60% [36]. Anisometropia was found among 7% of the group at ≥1.00 D in a spherical equivalent in the study by Ikeda et al. [19]. It is worth noting that an eye examination of a child with ASD is incredibly difficult and can come with false refraction measurements due to the challenging nature of cooperation between the examiner and an autistic patient.

### 4.2. Fundoscopy

Fundoscopies and morphologies of the optic nerve disc are two of the most routinely conducted eye examinations. Isolated reports in the literature can be found on this topic. For example, Denis et al. observed optic disc pallor among 4 (40%) of 10 examined children [36]. Pale optic discs in children with ASD are usually associated with atrophy or hypoplasia of second cranial nerve (CN II). The most likely factors leading to optic nerve atrophy in the pediatric group in general include brain tumors, trauma, neurodegenerative diseases, vitamin deficiencies (biotin, B12) in nursing mothers, and post-infectious and perinatal complications. Hypoplasia factors, on the other hand, consist of complications related to endocrine and nervous system disorders [37]. In the study by Chang et al., the most common fundus lesion noted was the neuropathy of the II nerve, which occurred in 4% of the children studied, secondary to hydrocephalus and associated with a congenital optic nerve anomaly [38]. Ezegwui et al. also showed changes in the fundus image in 18% of the cases studied (temporal optic disc pallor and bilateral maculopathy with diffuse choroidal atrophy) [35]. A further cause of the atrophy of CN II is the deficiency of vitamin B12, which may be preceded by a gradual decrease in visual acuity [39]. Vitamin deficiencies are a frequently observed phenomenon in children with ASD due to the adherence to selective diets resulting from behavioral disorders and sensory sensitivities, in which children eat only color-selected foods [40]. In contrast to the aforementioned ideas, there are contrary articles in literature that do not indicate any deviation from normal fundus detail [18,34,41,42,43].

### 4.3. Optical Coherence Tomography (OCT) in Children with ASD

The retina, being an extension of the neurons of the central nervous system, is the outermost and most easily accessible layer of nerve cells used for the diagnosis of neuro-ophthalmic diseases. Currently, OCT examination is becoming a popular additional technique to help diagnose neurodegenerative diseases, i.e.,: multiple sclerosis, schizophrenia, Alzheimer’s disease, and Parkinson’s disease, where retinal differences in the macula and RNFL are found in the image of the second cranial nerve (retina nerve fiber layer—RNFL) disc [44]. OCT examination of the macula is a rarely described test used in children with ASD. One of the first was a study by Emberti Gialloreti et al. that showed a correlation between the overall retinal thinning and a lower verbal IQ, which is responsible for the abstract thinking skills in these children [45]. In addition, this study was also one of the first to show the possibility of using changes in the thickness of the retinal nerve fiber layer in the OCT examination in 2013 [45].

Contradictory results were obtained by Garcia-Medina et al., who showed an increase in the total retinal thickness within the macula due to the thickening of the inner nuclear, plexiform layers, inner nuclear layer, and the perinuclear layer, and an increased thickness of the nerve fiber layers in the inferior and nasal quadrants and the temporal and inferior sectors of the RNFL in children with ASD compared to healthy children [46]. In a later study, the same author showed structural and vascular changes within the retina (including an increase in central retinal subfield thickness: 266.92 μm) in children with autism compared to healthy subjects. It has been hypothesized that an increased retinal thickness in children with ASD correlates with neuroinflammation or the presence of vascular anomalies [47]. The most frequently mentioned vascular changes are the disruption of the blood–brain barrier (most notable is pathology within the tight junction protein claudin-5) [48], inflammation within the vascular endothelium, and reduced angiogenesis, leading to decreased vasculature within the brain, which results in hypoxia [49]. Published in 2016, a study by Little et al. showed no significant difference in the thickness of individual retinal layers between children with ASD and healthy children [50]. However, the results of the latest research prove the practicality of OCT as a diagnostic tool in children with autism. In the Kara et al. study presented at the 11th International Congress on Psychopharmacology and 7th International Symposium on Child and Adolescent Psychopharmacology, it was shown that in people with ASD, significant differences are only seen in some locations, such as the RNFL sublayers, choroid, ganglion cell layer (GCL), and inner plexiform layer (IPL). The left nasal quadrant turned out to be a particular area of thinning. This result may indicate progressive degeneration over the course of ASD [51]. What is more, the 2022 study by Bozkurt et al. showed a significantly lower thickness in a larger proportion of the retina (temporal, temporal superior, nasal superior, temporal superior, and global RNFL) in the ASD group than in the healthy children. However, no correlation was found between the RNFL thickness and ASD symptom severity. The study also put forward a thesis on the possible link between the thinning of this layer and the current neurodevelopmental changes in children with ASD [52].

Neuroinflammation is characterized by the persistent activity and proliferation of glial cells (astrocytes and microglia) and their increased release of inflammatory mediators (i.e., cytokines, chemokines: interferon-γ (IFN-γ), interleukin (IL)-1β, IL-6, IL-12p40, tumor necrosis factor-α TNF- α, and CC-chemokine ligand 2 (CCL2)) in the cerebral cortex—particularly in the frontal lobe, cerebellum, and cerebrospinal fluid [53,54]. Their subsequent effect on changes in the CNS occurs mainly through the regulation of neuronal function and connectivity, which manifests as cognitive impairment in these children [53]. Furthermore, there are reports of differential functions of microglia, which have been divided into M1 microglia, which promote neuroinflammation and subsequent neuronal dysfunction, and M2 microglia, which have an anti-inflammatory effect, leading to homeostasis. Thus, a theory has been advanced that there is a disturbed balance between M1 and M2 and a greater predisposition to pro-inflammatory activity in children with ASD [54]. 

Many children with ASD also struggle with other neurological diseases, such as epilepsy [25]. A study by Bilen et al. on the comparison of retinal thickness between healthy subjects and those with epilepsy showed significant thinning of this layer in patients, a different result than in the OCT images of the retinas of autistic children. This finding confirms generalized neurodegeneration in patients with epilepsy and may prove to be a useful tool for tracking and assessing neurodegenerative changes in ASD [44]. Ophthalmic symptoms regarding both fundus and retinal images from OCT examinations may precede the onset of neurological symptoms, further confirming the close correlation between the retina and the brain [55]. 

### 4.4. Electrophysiological Studies

Another ophthalmologic test that can be used in the diagnosis of ASD is electroretinography (ERG). The ERG test result indicates an association with neural pathways, neurotransmitters, and their receptors [56]. It is used to record the electrical potential evoked by light coming from the retina in response to a light stimulus under both scotopic conditions for rods and photopic conditions for the functional role of the cone photoreceptors. One of the first studies by Ritivo et al., dating back to 1988, showed a reduction in the b-wave amplitude in children with autism. It was concluded that this finding may be correlated with autism or developmental disabilities in children with ASD. It is also interesting to observe a loss of Purkinje cells in patients with ASD [57], which, in agreement with the findings of Castrogiovanni and Marazziti, is associated with dysfunction within the dopaminergic system, which plays a role in relaying in the inner nuclear layer. It has been shown that a change in the b-wave amplitude recording can be useful in diagnosing dopaminergic disorders, both within the retina and in the CNS [58]. This is confirmed by a study by Lavoie et al., which showed abnormal ERG results for rod photoreceptors among 48% of children with ASD, i.e., a decreased b-wave amplitude under scotopic stimulation with a dim blue and red light [59]. The results of Constable et al. are similar, which showed a reduction in the b-wave amplitude under photopic conditions along with atypical bipolar cell function. In addition, b-wave reduction with prolonged flashing (photopic conditions—“ON,” not “OFF”) has been demonstrated in individuals with ASD. Some individuals with ASD also show subnormal DA (dark-adapted) b-wave amplitudes [56]. These exploratory findings suggest that there is an altered bipolar cone-ON signaling in ASD.

Another study conducted by the same author examined a further part of the electroretinogram recordings called the photopic negative response (PhNR, which has not yet been evaluated in ASD). The PhNR evaluates the function of the proximal retina: mainly ganglion cells, amacrine cells, and glial cells. The results of the aforementioned studies indicate that the retinal ganglion cell function is not altered in children with ASD [60]. 

In summary of both the earlier [56] and present results, it was concluded that the normal PhNR image along with the reduced b-wave amplitude in ERG examination in children with ASD may be related to altered synaptogenesis between photoreceptor and bipolar cells [60]. Similar results were obtained in the 2022 study by Lee et al., in which the ERG b-wave amplitude was recognized as a characteristic differential feature for ASD, which generated the b-wave response amplifier. These circuits occur in the brake neurotransmission between glutamate and GABA, which are primarily responsible for the production of the b-wave and may be useful as a biomarker for ASD neurodevelopment [61]. A different thesis was adopted by Friedel et al., which showed no significant difference in the ERG results between children with autism and the control group [62]. An interesting take was described by Manjur et al., who investigated the classification and probability of the ASD occurrence based on the introduction of the ERG. The method showed 86% accuracy, which exceeds the previous methods (time-domain features) with 65% accuracy [63].

## 5. Conclusions

This article is a review of the current possibilities of early diagnosis of neuroinflammatory changes in the brain through the use of ophthalmological tests, such as OCT and ERG, which correlate with changes in both the thickness of the anatomical structures of the retina in the macula and the thickness of the optic nerve fibers, as well as with the functional changes of retinal photoreceptors. It seems important that the ERG test results indicate an association with neural pathways, neurotransmitters, and their receptors. Therefore, the normal PhNR image along with the reduced b-wave amplitude in ERG examinations in children with ASD may be related to altered synaptogenesis between photoreceptor and bipolar cells. Nowadays, OCT examination is also becoming a popular additional technique to help diagnose neurodegenerative diseases, as well as ASD. Hence, OCT examination of the macula can be used as an early marker of neuroinflammatory changes in the brain, which correlates with a lower verbal IQ and is responsible for abstract thinking skills in these children. In addition, this study may also be one of the first to show the possibility of using changes in the thickness of the retinal nerve fiber layer in the OCT examination. Despite the large amount of research, the etiopathogenesis of the development of autism in children still remains unknown. Interestingly, it remains possible to diagnose retinal changes in relation to CNS changes among patients with ASD. Comprehensive ophthalmic evaluations, refractive assessments, and the consideration of functional changes in photoreceptor cells, and disorders of retinal structures or optic nerve fibers confirmed by OCT and ERG studies may become auxiliary diagnostic tools for the early detection of autism in children with ASD in the future.

## Data Availability

No new data were created or analyzed in this review. Data sharing does not apply to this article.

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
