# Peer review of "The Use of Optical Coherence Tomography and Electrophysiological Tests in the Early Diagnosis of Inflammatory Changes in the CNS in children with ASD—A Review of Contemporary Literature"

_ijerph, 2023, doi:10.3390/ijerph20043591_

Round 1

Reviewer 1 Report

The manuscript entitled, Manifestation of changes in the organ of vision and use of OCT and ERG tests for early diagnosis of neurological changes in children with ASD in relation to the risk factors – a review of contemporary literature, by the authors, Modrzejewska and Bosy-GÄ…sior provides a review of literature from 2017 on various factors that are related to neurological pathogenesis and finally concluding that ophthalmic screening especially, the OCT and ERG screening could become a diagnostic tool for early detection of ASD.

Suggested comments: 

1. The manuscript is written as a generalized review. In order to make it more relevant to the title of the manuscript, I would suggest the authors rearrange the format of the manuscript. To justify the title, section 4: Congenital and Perinatal factors with the subtitle 4.1: Perinatal disorders and CNS damage, be moved after section 2: ASD biological substrate, for continuity. Rearranging the section would be more meaningful to bring the reader’s focus to the significance of the title of the manuscript. 

2. The conclusion should reiterate more affirmatively the purpose of their title, delineating the importance of OCT and ERG screening, discussing the limitations, and suggesting that more data would substantially improve our understanding as there are not many studies performed. This manuscript can provide the necessity or awareness that the vision test could serve as an auxiliary diagnostic tool. 

3. Since this is a review, I would suggest the authors cite the relevant literature on the factual statements. Some examples:  Pg1 ln 30-33 with the appropriate reference where the percentages were derived from each geographical location.  Similarly, the sentence in Pg1 ln38 mentioning the influence of cytokines IL-17A and IL-2 linked to encephalitis needs appropriate citation. And, also, Pg4 ln194-196 needs citation. 

4. Pg4 ln196-198, the sentence is vague and does not convey the intended message, it needs to be revised.

5. Having given the total number of manuscripts reviewed, I would suggest the authors to please read carefully the ambiguous sentences and cite the relevant literature for the factual statements. 

Author Response

 Responses to Reviewer 1 

Special thanks to the reviewers for valuable tips and accurate indication of the weak points of our work. We are glad that the article aroused interest - which allowed us to take a closer look at our work and supplement it by expanding the literature with the indicated authors and indicated topics.

  1. At the request of the reviewer, the order of fragments of the text 4: Congenital and Perinatal factors with the subtitle 4.1: Perinatal disorders and CNS damage, is now moved after section 2: ASD biological substrate to make the topic coherent and understandable for the reader.

  2. The conclusion was changed in order to better summarize the topic and present the importancy of OCT and ERG screening. The authors also tried to prove that the above-mentioned diagnostic tests may turn out to be future-proof in the early detection of autism in children.

  3. Missing citations were added in places designated by the reviewer.

  4. The sentence has been deleted. The authors stated that the sentence is not needed in the manuscript because it does not contain essential information.

  5. The authors have made every effort to improve the manuscript in accordance with any comments from the reviewer. For which we want to thank.

The authors would also like to add that the grammar in the article has been checked and corrected. The authors are waiting for further comments, we will be grateful - for pointing out more points that we can change so that the article is well received by readers.

Once again thank you for your review and we look forward to further guidance

Kind regards,

Monika Modrzejewska

Reviewer 2 Report

Reviewer suggestions.

This is a a really interesting article and could make a useful consolidation of ASD and diagnostic vision research for readers. 

1. Title needs to be shorter and changed [Manifestation of changes in the organ of vision and use of OCT and ERG tests for early diagnosis of neurological changes in children with ASD in relation to risk factors – a review of contemporary literature]

- What are the key risk factors, i.e neuroinflammation should be stated in title, if this is the key topic for risk.

- If the title says ‘use of OCT and ERG’ then there needs to be more literature describing these methods in the main body of the article.

2. The specific 'risk factors' need to be clearer in the main body of the review to reflect the title

3. Search strategy: The article states 'Due to the limited amount of articles on visual impairment among children with autism (Pub- 51 Med search engine: 44 articles)'. 

There are many ASD vision studies attempting to link the autism with underlying visual processing. A pubmed search retrieved 684 articles, using the general search term: 'visual impairment among children with autism' (although these are not confirmed for relevance). It would be good to report the specific search string used that retrieved the numbers of articles for the authors, and what type of articles were excluded to reach a number of 44.  

I did a quick check on use of retinal thickness in the diagnosis of autism using the OCT method, and there is an article  missed for inclusion/ discussion:

Little et al., 2016: study 'captured OCT images of line and volume scans in 29 children with ASD and age-matched controls, and did not find any difference in overall retinal thickness in the area surrounding the fovea. However, on layer segmentation analysis there was thinning in the inner retinal layers approaching the optic nerve head in the ASD group'.

Little J-A, Anketell PM, Doyle L et al. Investigation of retinal thickness using OCT in autism spectrum disorder. Invest Ophthalmol Vis Sci 2016; 57: 12. ARVO E-abstract 3100.

It would be good to rerun the pubmed searches in case other articles can be included in the review, especially for the OCT and ERG methods.

4. Formatting: the size of the text is not consistent throughout this review.

I hope these suggestions are helpful for your manuscript.

Author Response

 Responses to Reviewer 2 

Special thanks to the reviewers for valuable tips and accurate indication of the weak points of our work. We are glad that the article aroused interest - which allowed us to take a closer look at our work and supplement it by expanding the literature with the indicated authors and indicated topics.

  1. The title has been changed as requested by the reviewer. The authors agree with the reviewer's opinion that at that time the title was too long and did not present the main point of the article.
  2. The authors in the title focused on the main problem of neuroinflammation caused by various risk factors. "Neuroinflammation" was placed in the title as common pathophysiological mechanism for the risk factors described in the article.
  3. Repeated search for the latest literature focused on oct and erg research - which indicated new items not yet noticed by the authors - which have already been included in the discussion, which is an important supplement to our text - for which we thank the reviewer.
  4. The authors have tried to format the text size to be consistent throughout the review.

The authors would also like to add that the grammar in the article has been checked and corrected. 

The authors are waiting for further comments, we will be grateful - for pointing out more points that we can change so that the article is well received by readers.

Once again thank you for your review and we look forward to further guidance

Kind regards,

Monika Modrzejewska

Round 2

Reviewer 2 Report

The manuscript is much improved in quality with the reorganisation of title, content, and especially the additional references sourced in the OCT and ERG field. Well done to the study authors for taking on board all the feedback.

A few small changes:

Title: remove the hypen in children: chil-dren 

Abstract: remove or update the word 'Nowadays' as it doesn't sound scientific.

Author Response

 Responses to Reviewer 2

First of all, we would like to thank for the positive reviews of our article. We are glad that we were able to correct them and improve the article according to the reviewer's comments.

  1. We removed the hypen in „children” word in the title.
  2. We removed the word "nowadays" from the abstract as recommended by the reviewer.

Once again thank you for your review.

Kind regards,

Monika Modrzejewska